# Efficiency of Public R&D Management Agency and Its Improvement toward Open Innovation

**Byung-Yong Hwang** [1,*], **Sung-Hun Park** [1] **and Dae-Cheol Kim** [2]

1   Korea Institute of S and T Evaluation and Planning, Seoul 06775, Korea; mabsosapsh@kistep.re.kr
2   School of Business, Hanyang University, Seoul 04763, Korea; dckim@hanyang.ac.kr
*   Correspondence: byhwang@kistep.re.kr

**Abstract:** R&D management agencies have been taking on key roles in the national R&D ecosystem. The purpose of this study is to suggest ways to improve the role of R&D management agencies for R&D planning and evaluation. It analyzed the regulatory system and relative efficiency of the current use of planning and evaluation cost. Data-collection sources included documents, surveys, and interviews with staff members in agencies responsible for national R&D management. By using DEA models, relative efficiency was measured. Based on the analysis results, we present suggestions for improvement in three areas: (a) establishment of institutional basis; (b) improvement of budget process; (c) improvement measures depending on the causes of inefficiency. Finally, future directions and the limits of this study are discussed.

**Keywords:** National Innovation System; public R&D management agency; Data Envelopment Analysis; R&D performance; open innovation

## 1. Introduction

As the budget for national R&D projects recently increased to KRW 24.2 trillion (as of 2020), the functions and roles of public R&D management agencies (hereinafter "R&D management agencies") have been recognized as more important. The Korean government under President Moon Jae-in raised the need for efficient and systematic support for R&D management agencies in the "National R&D Innovation Act" [1]. The laws and systems related to R&D management agencies are as follows.

First, according to Article 23 (1) of the Act, the Minister of Science and ICT may conduct a fact-finding surveys and analysis of the designation and operation of a R&D management agency. Second, In Article 23 (2) of the Act, the Minister of Science and ICT may demand the designation or cancellation of the designation of a R&D management agency, improvement of its operational efficiency, etc. Third, a R&D management agency is a body designated by a central administrative organization to be entrusted with planning, evaluation, and management of R&D projects in accordance with Article 2, Paragraph 4 of the "National R&D Innovation Act" [1].

Meanwhile, there are currently 17 agencies under 13 ministries that manage national R&D projects, but a consistent management system related to planning, evaluation, and management is insufficient. As of 2020, R&D management agencies were carrying out planning, evaluation, and management of major R&D projects worth about KRW 8.5 trillion, and the ratio of the planning and evaluation costs to the total project cost was 3.1%. Nevertheless, support should be strengthened for R&D management agencies to be equipped with capabilities as a professional R&D project management agency and to act as a bridge between ministries and researchers.

However, it is also true that problems of insufficient transparency and efficiency concerning management of planning and evaluation costs are constantly being pointed out. Specifically, the National Assembly of the Republic of Korea is demanding R&D management agencies to improve operational efficiency through active implementation

of fact-finding surveys [2]. Baek et al. (2013) [3] pointed out that there is a problem in that ministries arbitrarily apply the rate set in the budgeting process due to the lack of accounting standards for planning, evaluation and management of national R&D projects.

Accordingly, there is always the possibility that the problem of under- or over-regulating the planning and evaluation management costs will occur, and from the point of view of R&D management agencies, inefficiencies may occur because the institution's operating costs cannot be reliably guaranteed. Song et al. (2015) [4] stated that when planning and evaluation expenses are supported at a uniform ratio of the government subsidy budget, the characteristics of the institution are not reflected, which may cause difficulties in operation and a decrease in the quality of overall research management.

Problems commonly raised in the preceding studies include under-/over-setting and lowering of operational efficiency as the planning and evaluation cost is determined irrespective of the cost structure. Existing research content presents problems based on budget and expenditure information for each R&D management agency prior to 2017. This study intends to analyze the current status and problems related to planning and evaluation costs in more detail from the perspective of efficiency based on 2020 data.

Meanwhile, from the viewpoint of R&D productivity, the concepts of R&D effectiveness and R&D efficiency are important. In Korea, the efficiency of output compared to input is important when selecting public R&D projects. In addition, the effects of achieving the research purpose are usually judged qualitatively, and autonomy is given to the research institution as much as possible. Accordingly, this study will focus on the efficiency of public R&D management agencies and the possibility of open innovation.

In other words, the purpose of this study is to suggest ways to improve the role of R&D management agencies for R&D planning and evaluation. It analyzes the regulatory system and relative efficiency of the current use of planning and evaluation cost. Specifically, this study focuses on three questions: first, is R&D management agencies' regulatory system of planning and evaluation cost efficient? Second, what are the variables that affect relative efficiency? Third, how can the role of R&D management agencies in R&D planning and evaluation be improved?

## 2. Research Background and Methodology

### 2.1. Overview of Public R&D Management Agencies

First, this paper aims to examine the functions and roles of R&D management agencies, which are the main actors of planning and evaluation costs. According to Hong et al. (2018) [5] and Hwang et al. (2019) [6], as shown in Figure 1, the management and coordination mechanism of national R&D projects can be divided into pan-ministries, science and technology-related ministries, supporting organizations, and industry-university-research institutes. Supporting organizations herein refer to R&D management agencies, which receive their R&D budget from the government, and manage and execute them.

In the field of science and technology policy, it is important that the research support mechanism naturally induces sectoral cooperation to achieve public objectives [7]. Developed countries promote the professionalism and efficiency of research funding support rather than direct government support with a mechanism in place to ensure autonomy and independence of research institutes and to overcome the bureaucratic culture in research. These institutions are quasi-governmental institutions between the government and research institutions, such as the Research Council, the Foundation, and R&D management agencies under each ministry, through which efforts should be made to ensure the independence and autonomy of those who carry out research through indirect support.

The functions and roles of R&D management agencies are as shown in Figure 2, as an intermediate body in the national R&D budget allocation system in the national R&D project management hierarchy, and they are responsible for planning, managing, and evaluating research projects and tasks that reflect the characteristics of the project by being entrusted with the R&D budget of the ministry, and supporting commercialization through performance management [8]. Looking at this in detail, R&D planning means setting the

direction and designing a new project through technology demand survey and long- and short-term forecasting. R&D management includes performance management activities such as signing project and task agreements, project promotion and progress management, research fund management, and research results utilization. R&D evaluation refers to various support activities such as task announcement, task selection and selection evaluation, and interim and result evaluation [9]. Furthermore, R&D management agencies should be in charge of performance management functions that can improve performance and increase investment efficiency throughout the R&D life-cycle while promoting other projects, such as technology transfer, international cooperation, and human resource training.

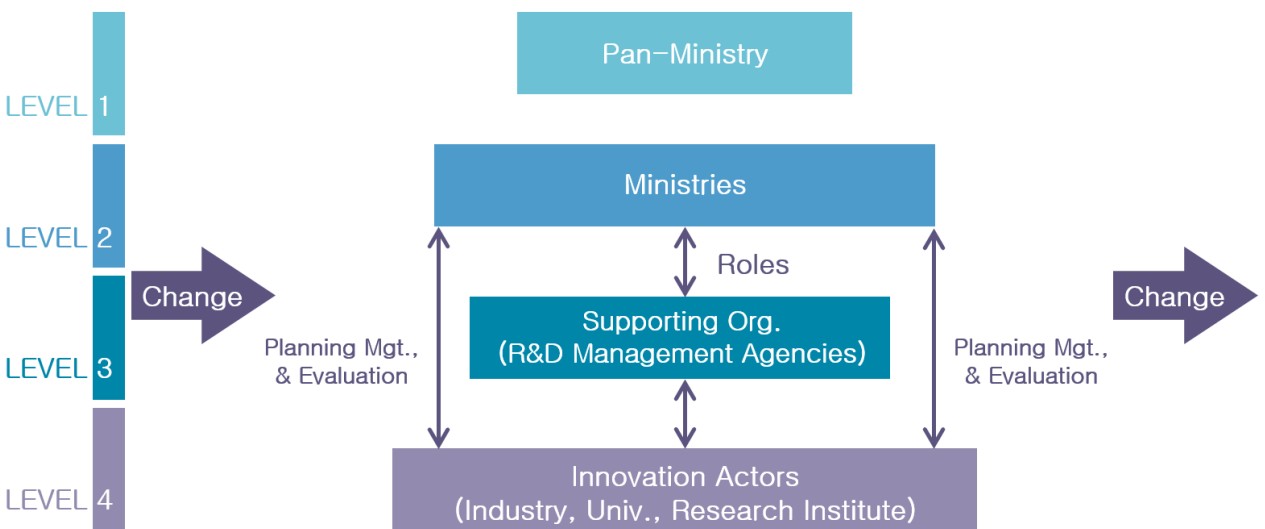

**Figure 1.** Mechanism of the national R&D program management in the National Innovation System. Source: The Impact of Organizational Competencies on the Performance of R&D Management Agencies in Korea. Hong et al. (2018) [5], Hwang et al. (2019) [6].

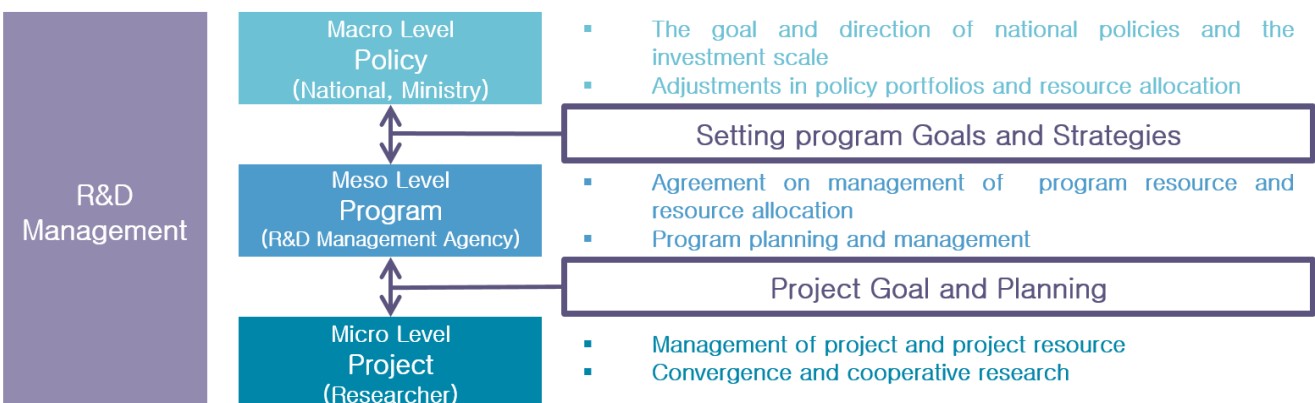

**Figure 2.** Levels of managing science and technology policies, programs, and projects. Source: The Impact of Organizational Competencies on the Performance of R&D Management Agencies in Korea. Hong et al. (2018) [5], Hwang et al. (2019) [6].

Second, the aim is to examine the operational status of R&D management agencies by type. Currently, 13 central administrative organizations are operating 17 R&D management agencies to entrust the planning, evaluation, and management of R&D projects under their jurisdiction. As shown in Table 1, 17 R&D management agencies can be classified into two types according to their R&D management duties.

**Table 1.** Current status of R&D management agencies (2020).

| Type | Number of Agencies | R&D Management Agency |
|---|---|---|
| Main Purpose (Management of R&D Projects) | 8 EA | National Research Foundation of Korea (NRF)<br>Institute for Information and Communications Technology Promotion (IITP)<br>Korea Evaluation Institute of Industrial Technology (KEIT)<br>Korea Institute of Energy Technology Evaluation and Planning (KETEP)<br>Korea Technology and Information Promotion Agency for SMEs (TIPA)<br>Korea Institute of Marine Science and Technology Promotion (KIMST)<br>Korea Agency for Infrastructure Technology Advancement (KAIA)<br>Korea Institute of Planning and Evaluation for Technology in Food, Agriculture and Forestry (IPET) |
| Supplementary Tasks (Management of R&D Projects) | 9 EA | Korea Institute for Advancement of Technology (KIAT)<br>Korea Health Industry Development Institute (KHIDI)<br>Korea Environmental Industry and Technology Institute (KEITI)<br>Korea Creative Content Agency (KOCCA)<br>Korea Copyright Commission<br>Korea Sports Promotion Foundation<br>Korea Meteorological Institute<br>Korea Foundation of Nuclear Safety<br>Korea Forestry Promotion Institute (KOFPI) |

Source: Hwang et al. (2021) [10].

Out of two types, eight agencies are mainly focused on R&D projects management, and nine consider R&D projects management as Supplementary task.

Third, planning and evaluation costs are defined as expenses required for R&D management agencies to carry out tasks such as planning, evaluation, management, and performance utilization promotion of each department's R&D project, and can be divided into broad and narrow terms [11]. In a broad concept, it refers to all expenses incurred in the planning, evaluation, and management of R&D projects entrusted by R&D management agencies while narrowly it can be seen as a budget excluding overhead expenses among expenses for planning, evaluation, and management of the entrusted R&D project [12–14].

In addition, the components of the planning and evaluation costs of R&D management agencies are usually composed of employment cost, direct cost, and indirect cost, and the direct cost accounts for largest portion among the components of each expense item. As a result of this survey, it was found that the ratio of planning and evaluation costs in the form of commissions on average 3.1%, ranging from the minimum of 0.6% to the maximum of 4.3%, depending on the detailed project.

### 2.2. Data Envelopment Analysis

The data envelopment analysis (DEA) model was used to analyze the efficiency of R&D agencies. DEA is an analysis technique mainly used to measure the efficiency of research subjects in various fields [15]. The DEA model, first proposed by Charnes, Cooper and Rhodes (1978) [16], is based on the concept of Farrell (1957) [17] of relative efficiency. This is a method of measuring the efficiency value of the analysis target by deriving the empirical efficiency frontier by considering multiple input and output factors at the same time, and then comparing it with other DMUs (decision-making units). Here, DMU refers to the object to be analyzed, and in this study, R&D agencies correspond to this.

The DEA model can be divided into the CCR [16] and BCC [18] models according to the assumption of the scale effect and be classified as an input- or output-based models according to the purpose of the efficiency measurement.

If there are n DMUs, and each DMUj (j = 1,2, … , n) has m inputs (i = 1,2, … , m) and s outputs (r = 1, 2, … , s), the input-based CCR model and the output-based CCR model are as shown in Equations (1) and (2) below. Unlike the CCR model, a constraint of $\sum_{j=1}^{n} \lambda_j = 1$ is added in the BCC model. This assumes variable scale returns and means the convexity of the BCC model.

Input-based CCR model:

$$\text{Min}\theta$$
$$\text{s.t. } \theta x_i \geq \sum_{j=1}^{n} x_{ij}\lambda_j \quad i = 1, 2, \dots, m,$$
$$y_r \leq \sum_{j=1}^{n} y_{rj}\lambda_j \quad r = 1, 2, \dots, s,$$
$$\lambda_j \geq 0, \quad j = 1, 2, \dots, n.$$
(1)

Output-based CCR model:

$$\text{Max}\varphi$$
$$\text{s.t. } x_i \geq \sum_{j=1}^{n} x_{ij}\lambda_j \quad i = 1, 2, \dots, m,$$
$$\varphi y_r \leq \sum_{j=1}^{n} y_{rj}\lambda_j \quad r = 1, 2, \dots, s,$$
$$\lambda_j \geq 0, \quad j = 1, 2, \dots, n.$$
(2)

Input-based BCC model:

$$\text{Min}\theta$$
$$\text{s.t. } \theta x_i \geq \sum_{j=1}^{n} x_{ij}\lambda_j \quad i = 1, 2, \dots, m,$$
$$y_r \leq \sum_{j=1}^{n} y_{rj}\lambda_j \quad r = 1, 2, \dots, s,$$
$$\sum_{j=1}^{n} \lambda_j = 1$$
$$\lambda_j \geq 0, \quad j = 1, 2, \dots, n.$$
(3)

Output-based BCC model:

$$\text{Max}\varphi$$
$$\text{s.t. } x_i \geq \sum_{j=1}^{n} x_{ij}\lambda_j \quad i = 1, 2, \dots, m,$$
$$\varphi y_r \leq \sum_{j=1}^{n} y_{rj}\lambda_j \quad r = 1, 2, \dots, s,$$
$$\sum_{j=1}^{n} \lambda_j = 1$$
$$\lambda_j \geq 0 \quad j = 1, 2, \dots, n.$$
(4)

On the other hand, the CCR model refers to the constant returns scale (CRS), that is, technical efficiency when it is optimal in terms of scale. Conversely, the BCC model refers to variable returns to scale (VRS), that is, technical efficiency when not optimal in terms of scale. Therefore, the inefficiency according to the scale can be inferred from the difference in technical efficiency of these two models. First, the scale efficiency (SE) value, which is the ratio of the CCR value to the BCC value, is calculated. Compared with the BCC value, the inefficiency in the lower side is considered to be greater. Here, the BCC value is called pure technical efficiency (PTE). Here, BCC value means pure technical efficiency (PTE) and CCR value means technical efficiency (TE). In addition, since the VRS model does not assume the optimal scale state, it is possible to determine whether or not the scale is profitable by using the lambda value in the CRS model. In other words, it is possible to find a way to increase efficiency by determining whether the current DMU is in a state of 'economy of scale' or a state of decreasing 'economy of scale'.

Meanwhile, the following studies applied DEA to the efficiency analysis of domestic public institutions. Lee (1993) [19] analyzed the relative efficiency of government-funded research institutions by applying DEA to suggest alternatives to the existing institutional evaluation. The study suggested that DEA analysis can provide an objective score of public institutions and that DEA is a method that can complement traditional evaluation methods. Park (2006) [20] measured the efficiency index and excess input of each branch through DEA analysis to compare and analyze the performance of 12 branches of the Korea Energy Management Corporation from 2000 to 2003. The trend of changes by year of the relative efficiency ranking was analyzed, and it was explained that the framework of performance measurement applied by the Korea Energy Management Corporation can be scientifically

supplemented through DEA. Yu (2009) [21] conducted a DEA analysis on the Jeonbuk Development Corporation to present an evaluation methodology that can evaluate the management efficiency of public institutions instead of the existing management evaluation. The study suggested that it is difficult to conduct effective efficiency evaluation with the current management evaluation system, and that a scientific evaluation system should be established through evaluation methodologies such as DEA.

Through various previous studies, it was found that DEA is a useful tool to measure the efficiency of public institutions. In this study, the DEA model was used to measure the efficiency of R&D management agencies, and congestion analysis was also conducted along with DEA. Here, congestion refers to a phenomenon in which one or more outputs increase when one or more inputs decrease [22]. In other words, when input decreases, output should also decrease, whereas congestion refers to a phenomenon in which output increases. It is mainly used to verify over-input factors. Since 'lax management' can be a major cause of the inefficiency of public institutions, this study aims to examine the efficiency of R&D agencies from various angles by using DEA and congestion analysis together.

### 2.3. Research Methods

This study intends to present possible ways to improve the operational efficiency of planning and evaluation costs of R&D management agencies.

To this end, first, it intends to analyze the structural problems by examining the regulatory system such as the current status of planning and evaluation cost of 17 R&D management agencies and related regulations from 2016 to 2020. For data collection, we carried out a survey from 1 April 2020 until 30 May 2020. In addition, this study referred to the results of an unstandardized interview held during the above survey period. Second, DEA analysis is used to measure the causes of relative efficiency and inefficiency of R&D management agencies, and to identify factors that appear to be over-input through the existence of congestion.

Meanwhile, input and output variables of R&D management agencies were selected for DEA analysis. In the case of R&D management agencies, it is difficult to set general R&D performance indicators, considering that most of the research outputs are the results of consigned management rather than the results of their own research. In consideration of these characteristics, Kim et al. (2011) [23] also selected variables in a real unit related to the operation of R&D management agencies. As input factors, manpower and planning and evaluation costs were selected, and as output factors, project cost and number of projects were selected. In this study, variables were selected based on the operational characteristics of R&D management agencies and prior research. As shown in Table 2 below, as input variables, manpower dedicated to planning, evaluation, and management were selected as labor input, and planning and evaluation cost as capital input, while project cost and number of projects are chosen as yields for output variables.

**Table 2.** Descriptive statistics by input and output variables of R&D management agencies (2020). (Unit: person, KRW million).

| Type | | Input Variables | | Output Variables | |
|---|---|---|---|---|---|
| Ministry | R&D Management Agency | Manpower (FTE) | Planning and Evaluation Cost | Project Cost | Number of Projects |
| Ministry of Science and ICT(MSIT) | NRF | 252 | 88,895 | 10,864,546 | 521 |
| | IITP | 229 | 48,070 | 1,087,592 | 124 |
| Ministry of Trade, Industry, and Energy (MOTIE) | KEIT | 257 | 86,090 | 2,038,812 | 190 |
| | KIAT | 182 | 85,860 | 1,840,144 | 310 |
| | KETEP | 123 | 28,594 | 772,757 | 196 |
| Ministry of SMEs and Startups (MSS) | TIPA | 138 | 48,814 | 1,142,590 | 76 |

**Table 2.** *Cont.*

| Type | | Input Variables | | Output Variables | |
|---|---|---|---|---|---|
| **Ministry** | **R&D Management Agency** | **Manpower (FTE)** | **Planning and Evaluation Cost** | **Project Cost** | **Number of Projects** |
| Ministry of Oceans and Fisheries (MOF) | KIMST | 55 | 17,762 | 356,728 | 90 |
| Ministry of Land, Infrastructure and Transport (MOLIT) | KAIA | 99 | 25,636 | 506,378 | 89 |
| Ministry of Health and Welfare (MOHW) | KHIDI | 86 | 9727 | 382,571 | 75 |
| Ministry of Culture, Sports, and Tourism (MCST) | KOCCA | 14 | 3824 | 69,457 | 12 |
| | Korea Copyright Commission | 2 | 202 | 8050 | 2 |
| | Korea Sports Promotion Foundation | 3 | 245 | 7637 | 5 |
| Ministry of Agriculture, Food, and Rural Affairs (MAFRA) | IPET | 32 | 8516 | 188,582 | 55 |
| Korea Forest Service | KOFPI | 6 | 820 | 28,340 | 15 |
| Ministry of Environment (ME) | KEITI | 69 | 12,724 | 238,542 | 70 |
| Korea Meteorological Administration | Korea Meteorological Institute | 16 | 1037 | 31,720 | 14 |
| Nuclear Safety and Security Commission(NSSC) | Korea Foundation of Nuclear Safety | 6 | 1602 | 33,549 | 3 |
| Mean | | 92.29 | 27,554.00 | 1,152,823.24 | 108.65 |
| SD | | 90.49 | 31,258.76 | 2,582,442.20 | 135.26 |
| Max | | 257.00 | 88,895.00 | 10,864,546.00 | 521.00 |
| Min | | 2.00 | 202.00 | 7637.00 | 2.00 |

FTE: full-time equivalent.

In addition, this study intends to conduct an input-based efficiency analysis considering that the output variable is the result of the previous year's performance and is confirmed at the beginning of the year due to the nature of a R&D management agency. Both the CCR model and the BCC model were used to infer inefficiency as well as efficiency values, and to analyze the profitability of scale.

## 3. Analysis of Regulatory System of Planning and Evaluation Cost

### 3.1. Current Status and Problems of Planning and Evaluation Cost

3.1.1. The Complexity of the Budget Structure

According to the survey, the income structure of the planning and evaluation cost are varied by a R&D management agencies—grants, planning and evaluation cost projects, planning and evaluation cost in the form of commissions within sub-projects, and outsourcing.

As shown in Table 3 below, there are classifications of income types—Type A (planning and evaluation cost in the form of commission) has 7 agencies, Type B (planning and evaluation cost in the form of grants + commission) has 6, Type C (planning and evaluation cost in the form of grants + commission + planning evaluation project) has 3 and Type D (grants + planning evaluation project) has one. Type A, which relies only on planning and

evaluation costs in the form of commissions within sub-projects, accounts for 41.2%, or seven agencies.

**Table 3.** Classification by Income type of planning and evaluation costs (2020).

| Type | Source of Planning & Evaluation Cost | R&D Management Agency | No. of Agencies (%) |
|---|---|---|---|
| A | Planning and evaluation costs in the form of commissions within sub-projects | TIPA | 7 (41.2%) |
| | | Korea Copyright Commission | |
| | | Korea Sports Promotion Foundation | |
| | | Korea Meteorological Institute | |
| | | KHIDI | |
| | | KIAT | |
| | | KOFPI | |
| B | Grants + planning and evaluation expense in the form of commission within detailed project | KEIT | 6 (35.3%) |
| | | KIMST | |
| | | KEITI | |
| | | KAIA | |
| | | IPET | |
| | | KETEP | |
| C | Grants + planning and evaluation cost in the form of commission within detailed project + planning evaluation cost project | NRF | 3 (17.6%) |
| | | IITP | |
| | | KOCCA | |
| D | Grants + planning evaluation cost project | Korea Foundation of Nuclear Safety | 1 (5.9%) |
| Sum | | | 17 (100%) |

In the case of a commission-type planning and evaluation cost (16 agencies) within a detailed project, it was found to be difficult to apply a uniform commission rate because it was determined in consideration of various income structures and project characteristics for each agency.

☆☆☆ Person in charge at △△△△△ Agency: The increase or decrease in the number of tasks within a project rather than the overall size of the project has a greater impact on the increase or decrease in the use of planning and evaluation cost. Nevertheless, the overall scale of the project is given priority when setting the planning and evaluation cost.

(Interview with the R&D management agencies in April, 2020)

☆☆☆ Person in charge at △△△△△ Agency: It is not fair to set the planning and evaluation cost by relative comparison between different bodies without considering various conditions and characteristics of each agency.

(Interview with the R&D management agencies in April, 2020)

The above interview results are seen as an example of pointing out the problem on how planning and evaluation cost is set.

3.1.2. The Insufficiency of the Planning and Diffusion of R&D Results

Table 4 shows that if we look at the proportion of planning and evaluation costs by purpose of use, it is found that planning and evaluation costs are focused on evaluation rather than on planning and result diffusion to prevent fairness and transparent management of research funds.

**Table 4.** Average ratio of major R&D planning and evaluation costs by purpose of use (2016–2020). (Unit: %).

| 17 Agencies Average Ratio | Planning | Evaluation | Performance Mgt. | Others | Total |
|---|---|---|---|---|---|
| 2016 | 25.0% | 51.8% | 12.7% | 10.5% | 100% |
| 2017 | 24.4% | 55.9% | 12.4% | 7.4% | 100% |
| 2018 | 27.9% | 53.8% | 11.2% | 7.0% | 100% |
| 2019 | 26.4% | 55.2% | 11.3% | 7.1% | 100% |
| 2020 | 26.8% | 56.7% | 10.0% | 6.5% | 100% |

Overall, spending on planning increased by 1.8% over the past five years, while evaluation cost increased by 4.9%. Specifically, as of 2020, the highest average of 56.7% was spent on evaluation. On the other hand, only 26.8% of the planning and evaluation cost is spent on new project planning, such as trend analysis, research, and idea generation related to R&D projects, and an average of 10.0% is spent on performance management.

According to the government R&D investment direction and criteria for 2021 [24], 246 projects worth KRW 139.4 billion were reflected as major new R&D projects in 2021, and all new projects should be submitted through a preliminary planning report in accordance with relevant laws and regulations. In principle, the budget was not reflected for projects that have no report submitted or incomplete according to the criteria arranged.

However, despite the increase in new projects following the expansion of government R&D investment and the obligatory submission of pre-planning reports for new projects, it is figured out that support for pre-planning is confined to a limited number of agencies. Also, in most cases, the planning cost for a new project is not reflected in the planning and evaluation cost of the R&D management agency, so it is inevitable to take from the allocated existing planning and evaluation cost. As a result, it is found that the decline in planning quality due to the lack of support for the pre-planning cost becomes one of the main causes of the decline in the overall R&D research performance.

☆☆☆ Person in charge at △△△△△ Agency: In the current government budget system, the planning cost of a new project is not reflected in the planning and evaluation cost of the current year, so it is difficult to implement it in the planning and evaluation cost of other projects under management.

(Interview with R&D management agencies in April, 2020)

The above interview result points out the problem of insufficient funding for planning and evaluation costs required for planning and performance management and execution.

3.1.3. Little Connection between the R&D Evaluation and the Funding System

The result also shows that there has been insufficient independent performance evaluation and budget adjustment process related to the planning, evaluation, and management of national R&D projects, which are the unique roles of R&D management agencies.

Table 5 points out that each ministry is supporting the planning, evaluation, and management by setting its own budget for planning and evaluation without linking with the performance of R&D management agencies' planning, evaluation, and management support activities. Specifically, it was found that when setting up the budget for planning and evaluation costs, consultations between 17 agencies and the relevant ministries were basic but for five agencies they also considered applicable regulations at the same time.

In addition, the Ministry of Strategy and Finance's management evaluation of public institutions [25] is being conducted every year focusing on whether the management objectives have been achieved, but the evaluation results are linked only to the performance pay of the head of an organization.

**Table 5.** Budgeting method for major R&D projects (2020).

| Ministry | R&D Management Agency | The Way the Planning and Evaluation Cost is Determined | |
|---|---|---|---|
| | | Consultation | Applicable Regulation |
| MSIT | NRF | O | |
| | IITP | O | |
| MOTIE | KEIT | O | |
| | KIAT | O | |
| | KETEP | O | |
| MSS | TIPA | O | |
| MOF | KIMST | O | O |
| MOLIT | KAIA | O | O |
| MOHW | KHIDI | O | |
| MCST | KOCCA | O | |
| | Korea Copyright Commission | O | |
| | Korea Sports Promotion Foundation | O | |
| MAFRA | IPET | O | O |
| Korea Forest Service | KOFPI | O | |
| ME | KEITI | O | |
| Korea Meteorological Administration | Korea Meteorological Institute | O | O |
| NSSC | Korea Foundation of Nuclear Safety | O | O |
| | Sum | 17EA | 5EA |

### 3.1.4. Not Enough Budget for Playing R&D Management Agencies Roles Fully

Through this survey, it was found difficult to predict the long-term planning, evaluation, and performance management because the planning and evaluation cost is highly variable depending on the change of the total project cost.

According to Table 6 as of 2020, 69.4% of planning and evaluation costs are financed from project evaluation expenses in the form of commissions within sub-projects. Accordingly, it is found that the planning and evaluation costs of R&D management agencies are strongly interlinked with the project scale in a year.

**Table 6.** Financial structure of planning and evaluation cost (2020). (Unit: KRW million, %).

| Grants | Planning and Evaluation Cost Project | Planning and Evaluation Cost in the Form of Commissions within Sub-Projects | Others * | Total |
|---|---|---|---|---|
| 61,586 (13.2%) | 71,292 (15.2%) | 325,198 (69.4%) | 10,342 (2.2%) | 468,418 (100.0%) |

\* Outsourcing, carryover, subsidy, etc.

Meanwhile, even though the budget for major R&D projects managed by R&D management agencies increased between 2016 and 2020, there were some cases in which the planning and evaluation costs in the form of commissions within sub-projects actually decreased. For example, although the management project budget of the KIAT increased an average of 4.4% per year, the planning and evaluation cost in the form of commission within sub-projects decreased by 3.5%. In addition, although the management project budget of the KHIDI increased an average of 3.0% per year, the plan and evaluation cost in the form of commission within sub-projects decreased by 5.2%.

☆☆☆ Person in charge at △△△△△ Agency: Due to the unstable financial structure since only the planning and evaluation cost is available only within sub-projects, there is a limit in accumulating professional management capabilities due to the use of non-regular workers when additional manpower is needed for project expansion.

(Interview with R&D management agencies in April, 2020)

In addition, the above interview results point out the instability of the planning and evaluation cost.

*3.2. Discussion*

Based on the analysis results of the regulatory system of planning and evaluation cost of the R&D management agencies, this paper proposes the following improvement measures.

3.2.1. Establishment of Institutional Basis

First, it is necessary to prepare an institutional basis for supporting planning and evaluation costs according to the purpose of use. To be specific, the relevant basis for the enforcement ordinance should be prepared related to the National R&D Innovation Act to make sure that the competent ministries request planning and evaluation costs according to the payment criteria, and the Ministry of Science and ICT review and adjust them.

In addition, the competent ministries need to prepare "general principles for payment criteria for planning and evaluation costs". As an example of the general principle, rigid expenses (employment costs, current expenses) and non-rigid expenses (planning/evaluation/performance management costs) are divided based on the existing actual execution budget for national R&D project management, which is reflected in the calculation of the planning evaluation cost for the next year in consideration of the inflation rate, etc.

Second, it is to propose the establishment of a monitoring system for the operation status of planning and evaluation costs of R&D management agencies. As shown in Table 7 below, it is necessary to relate the results of the survey on the operation of R&D management agencies to the budget for planning and evaluation costs.

**Table 7.** Monitoring system for planning and evaluation cost in R&D management agency.

| R&D Management Agency | | Ministry of Science and ICT | | Ministry of Science and ICT |
|---|---|---|---|---|
| Register spending status to NTIS per purpose | ⇨ | Understand the operating status of planning and evaluation costs via NTIS | ⇨ | Reflect in the budget for planning and evaluation of the next year |

In order to monitor the budget and settlement of planning and evaluation cost for each project, the National Science and Technology Information Service (NTIS) establishes a monitoring system for planning and evaluation cost and conducts fact-finding inspection based on objective data. Specifically, the current status of expenses for each purpose of use, such as planning, evaluation, and management cost, of sub-projects managed by R&D management agencies is registered in NTIS through which Ministry of Science and ICT understands the actual use. If there is a project management group within sub-projects, the project management group separately manages the status of spending on planning and evaluation for each purpose of use.

In addition, an annual survey and analysis of the operation of R&D management agencies such as R&D project management efficiency, planning and performance management efficiency, etc. should be conducted and linked with budget support for planning and evaluation cost.

3.2.2. Betterment of Budget Process

First, it is necessary to expand and support manpower for planning and performance management. Considering the simplification of the evaluation framework using

the Integrated R&D Information System (IRIS), it is necessary to redeploy the evaluation manpower within the organization and expand the surplus manpower to support planning and performance management.

In addition, it is necessary to redeploy simple management manpower saved through outsourcing, etc. to fields that require expertise such as research planning and performance management. This will enable R&D management agencies to focus more on creative planning and performance management functions based on expertise, taking the time and cost spent on simple repetitive tasks.

Second, it is necessary to newly establish a separate planning and evaluation cost project for R&D management agencies. For example, as shown in item 3 in Table 8 below, the "independent planning and evaluation cost project" will be newly established and supported for R&D management agencies that perform major R&D projects on behalf of each ministry.

**Table 8.** Pros and cons of each alternative to reorganizing the budget process for planning and evaluation costs.

| | Measure | Pros | Cons |
|---|---|---|---|
| P1 | Maintain status quo | The current planning and evaluation cost structure is a product of a long-term adjustment. | Insufficient expertise improvement in planning and performance management |
| P2 | Convert funding to grants fully | Beneficial for securing long-term stability of agency operation | Lack of flexible response to changes in projects subject to management |
| P3 | Establish "independent planning and evaluation cost project" and dualize from grants | Plan new project and enhance performance management Possible to do long-term planning, evaluation, and performance management and pan-ministerial planning and R&D relay | Considering the current complex budget process including grants, it is difficult to create the dualization in a short period of time |

Through this, it will be possible to separately support the pre-planning cost required for new project identification and planning, which could not be used in the existing planning and evaluation cost, and the management cost for strengthening follow-up management such as the spread of R&D performance after the project is completed. In addition, it is possible to support the cost required for linking with other ministries and planning multi-ministerial projects.

## 4. Analysis of Relative Efficiency of Planning and Evaluation Cost

### 4.1. Efficiency Analysis Results

As shown in Table 9, the efficiency analysis of R&D management agencies showed that 6 of the 17 institutions, including NRF, KETEP, and KHIDI, found to be efficient (PTE = 1.00). Furthermore, 11 agencies such as IITP, KEIT and KIAT showed relatively inefficiencies.

The overall efficiency mean was 0.72, and the standard deviation was 0.27, but 6 out of 11 inefficient agencies were less than 0.5, showing a large gap between agencies. The cause of inefficiency was analyzed to be higher due to pure technical efficiency than by scale efficiency (SE = 35.71%, PTE = 64.29%). On the other hand, in the 'Returns to scale' column, it is possible to determine whether or not the economy of scale is economical by dividing into CRS, DRS, and IRS according to the value of $\sum_{j=1}^{n} \lambda_j = 1$.

First, three agencies were found in the CRS, where the scale was optimal, including NRF, Korea Sports Promotion Foundation and KOFPI. Three agencies with increasing returns on scale (IRS) showed a higher rate of increase in revenue than the rate of increase in size, while 11 agencies had diminishing returns on scale (DRS). In other words, the ratio of agencies in the diseconomies of scale among the analysis subjects showed a rather high result.

**Table 9.** Efficiency analysis results of R&D agencies.

| DMU | | TE (CCR) | PTE (BCC) | SE (TE/PTE) | Cause of Inefficiency | Returns to Scale |
|---|---|---|---|---|---|---|
| Ministry | R&D Management Agency | | | | | |
| MSIT | NRF | 1.00 | 1.00 | 1.00 | - | CRS |
| | IITP | 0.25 | 0.38 | 0.66 | PTE | DRS |
| MOTIE | KEIT | 0.32 | 0.36 | 0.90 | PTE | DRS |
| | KIAT | 0.71 | 0.82 | 0.87 | PTE | DRS |
| | KETEP | 0.65 | 1.00 | 0.65 | SE | DRS |
| MSS | TIPA | 0.25 | 0.26 | 0.98 | PTE | DRS |
| MOF | KIMST | 0.67 | 0.78 | 0.86 | PTE | DRS |
| MOLIT | KAIA | 0.37 | 0.50 | 0.75 | PTE | DRS |
| MOHW | KHIDI | 0.55 | 1.00 | 0.55 | SE | DRS |
| MCST | KOCCA | 0.36 | 0.39 | 0.91 | PTE | IRS |
| | Korea Copyright Commission | 0.64 | 1.00 | 0.64 | SE | IRS |
| | Korea Sports Promotion Foundation | 1.00 | 1.00 | 1.00 | - | CRS |
| MAFRA | IPET | 0.70 | 0.88 | 0.80 | SE | DRS |
| Korea Forest Service | KOFPI | 1.00 | 1.00 | 1.00 | - | CRS |
| ME | KEITI | 0.41 | 0.72 | 0.57 | SE | DRS |
| Korea Meteorological Administration | Korea Meteorological Institute | 0.72 | 0.77 | 0.94 | PTE | DRS |
| NSSC | Korea Foundation of Nuclear Safety | 0.22 | 0.42 | 0.52 | PTE | IRS |
| Mean | | 0.58 | 0.72 | 0.80 | | |
| SD | | 0.26 | 0.27 | 0.17 | PTE: 9 SE: 5 | CRS: 3 DRS: 11 IRS: 3 |
| Max | | 1.00 | 1.00 | 1.00 | | |
| Min | | 0.22 | 0.26 | 0.52 | | |

*4.2. Congestion Analysis Results*

In this study, congestion analysis was conducted to examine the efficiency of R&D agencies from various angles, and the results are shown in Table 10 below.

Among input factors, congestion in manpower occurred in three agencies, while congestion in budget showed quite high results. Nearly 60% of all agencies had congestion, and the scale was about 16.50%.

*4.3. Discussion*

4.3.1. DEA as the Tool of Analysis of the R&D Efficiency

DEA analysis measured the relative efficiency of R&D management agencies by using a model with manpower and planning and evaluation cost as input factors and project cost and number of projects as output factors, and suggested the causes of inefficiency and efficiency targets.

**Table 10.** Congestion analysis results of R&D agencies.

| DMU | | Efficiency (BCC) | Congestion (%) | |
|---|---|---|---|---|
| Ministry | R&D Management Agency | | Manpower (FTE) | Planning and evaluation Cost |
| MSIT | NRF | 1.00 | | |
| | IITP | 0.38 | | |
| MOTIE | KEIT | 0.36 | | 8.46 |
| | KIAT | 0.82 | | 42.50 |
| | KETEP | 1.00 | | 19.11 |
| MSS | TIPA | 0.26 | | 4.20 |
| MOF | KIMST | 0.78 | | 32.83 |
| MOLIT | KAIA | 0.50 | | 10.42 |
| MOHW | KHIDI | 1.00 | 4.67 | |
| MCST | KOCCA | 0.39 | | 11.12 |
| | Korea Copyright Commission | 1.00 | 5.58 | |
| | Korea Sports Promotion Foundation | 1.00 | | |
| MAFRA | IPET | 0.88 | | 28.53 |
| Korea Forest Service | KOFPI | 1.00 | | |
| ME | KEITI | 0.72 | | 6.08 |
| Korea Meteorological Administration | Korea Meteorological Institute | 0.77 | 20.31 | |
| NSSC | Korea Foundation of Nuclear Safety | 0.42 | | 1.74 |
| Mean | | 0.72 | 10.19 | 16.50 |
| SD | | 0.27 | 7.17 | 13.05 |
| Occurrence rate | | | 17.65 | 58.82 |

As a result of the analysis, first, out of the 17 R&D agencies, 6 were found to be efficient and 11 were found to be inefficient. The cause of inefficiency was found to be higher in pure technical efficiency than in scale efficiency. Second, as a result of the analysis of economies of scale, 11 agencies out of all agencies were found to be in a state of DRS, in which the rate of increase in revenue was lower than the rate of increase in size. Third, as a result of congestion analysis, it was found that among the input factors, the frequency and scale of congestion were higher in the planning and evaluation cost than in the manpower.

### 4.3.2. The Role of Public R&D Agencies as the Engine of Open Innovation Found from DEA Analysis

The findings of recent studies are as follows. By using the DEA models from the perspective of open innovation, Hsua and Hsueh (2009) [26] found that small enterprises conducting government-funded R&D projects showed higher efficiency. Park et al. (2011) [27] assessed the efficiency of R&D projects supported by the government in scientific and technological areas such as papers and patents by utilizing science and technology investment and performance data of NTIS (National Science and Technology Information Service). Hwang et al. (2019) [15] suggested improving the efficiency of public R&D through the improvement of the royalty system. Hwang et al. (2019) [6] suggested improving the pro-

ductivity of public R&D by improving the operational-efficiency of public R&D agencies. In addition, Yun and Zhao (2020) [28] conducted a study on business model development from the perspective of open innovation.

Based on the above related research and DEA analysis results, this study intends to discuss the improvement direction of the role of public R&D management agency as the engine of open innovation as follows. Nine agencies such as IITP, KEIT and KIAT, which showed lower pure technical efficiency than scale efficiency, should seek various ways to improve such as operational system improvement and organizational restructuring.

On the other hand, it was found that the cause of inefficiency of five agencies, such as KETEP and KHIDI was not due to pure technical efficiency, but rather to inefficiency of scale. This is thought to be able to reduce inefficiencies in the CCR model through appropriate management of manpower and planning and evaluation cost when outputs such as project cost and number of projects are given.

Rather than a strategy to increase output by expanding inputs, it seems desirable for these agencies to choose a strategy to reduce inputs while maintaining existing outputs. Such agencies can be interpreted as the need to reduce planning and evaluation costs. If it is difficult to reduce the planning and evaluation cost itself due to the operating conditions, it seems necessary to arrange the use of the current evaluation-centered planning and evaluation cost in the planning and performance sectors.

## 5. Conclusions and Further Research

This study analyzed the regulatory system and relative efficiency of the current use of planning and evaluation costs of R&D management agencies, and suggested improvement measures for operational efficiency. In this regard, this paper is meaningful as a 'country Report' that introduces public R&D management agencies in Korea. In addition, the limits of this study and further research directions are discussed.

As a result of the analysis of the regulatory system of the planning and evaluation costs of R&D management agencies, we identified the complexity of the budget structure, the insufficiency of the planning and diffusion of R&D results, little connection between the R&D evaluation and the funding system, and the insufficient budget for fulfilling R&D management agencies' roles.

In addition, as a result of measuring causes of relative efficiency and inefficiency through DEA analysis, and identifying factors that appear to be over-input, 6 out of 17 R&D management agencies were found to be efficient and 11 agencies were found to be inefficient, and inefficiency was caused more by purely technical efficiency than by scale efficiency. As a result of analysis on economies of scale, 11 out of all agencies were in a state of DRS. Congestion occurred in the majority of agencies.

Based on the above analysis results, the authors presented suggestions for improvement: (a) the establishment of an institutional basis; (b) improving the budget process.

In addition, the agencies whose inefficiencies are caused by pure technical efficiency should seek various improvement measures such as operating system improvement and organizational restructuring. It was also suggested that the agencies due to inefficiency of scale need proper management of manpower and planning and evaluation costs. Agencies in DRS status were suggested to choose a strategy to reduce inputs while maintaining existing outputs. It was found that institutions that showed congestion in the budget should reduce their planning and evaluation costs.

Finally, what is needed with regard to limitations and further research directions includes, first, a more systematic investigation in the process of investigating the current status of R&D management agencies since the conditions and research methods are different by agencies.

Second, the analysis was conducted with the target period between 2016 and 2020, but DEA analysis, in particular, should develop sophisticated models and indicators by reflecting the characteristics of the agencies.

**Author Contributions:** B.-Y.H. developed the concept and wrote the paper. S.-H.P. collected the data and performed the statistical analyses. D.-C.K. designed the research and wrote the paper. All authors have read and agreed to the published version of the manuscript.

**Funding:** This research was supported by KISTEP (Korea Institute of S&T Evaluation and Planning) (project number: AN21040).

**Institutional Review Board Statement:** Not applicable.

**Informed Consent Statement:** Not applicable.

**Data Availability Statement:** The data presented in this study are available on request from the corresponding author.

**Conflicts of Interest:** The authors declare no conflict of interest.

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
