# Peer review of "Efficiency of Public R&D Management Agency and Its Improvement toward Open Innovation"

_2199-8531, doi:10.3390/joitmc7030200_

Round 1

Reviewer 1 Report

The study explores the efficiency of R&D management agencies in South Korea. The following comments could improve the study.

In the introduction, paragraphs should not have just one sentence.

In the introduction, contributions of the study should be identified and briefly elaborated.

Line 58 has an incomplete sentence - According to Hong et al. (2018).

R and D should be R&D, which is a common terminology in the literature.

Which DEA method is used in the study – CCR or BCC and why?

What do the following abbreviations mean - SE, TE and PTE in Table 9?

In the conclusion, implications of the study should be elaborated in more details.

Author Response

  • Based on the comments of the judges, the paragraph has been modified in general so that it does not consist of a single sentence.
  • In the introduction part, the purpose and contribution of the study were added and revised in more detail. [Edit Line 54-59]
  • Line 58 sentence has been changed to Line 70 and the contents have been modified as follows. (→ Hong et al. (2018) [5] and…)
  • It has been modified to 'R&D' in accordance with the general terminology of the literature.
  • Both the CCR model and the BCC model were used to infer inefficiency as well as efficiency values, and to analyze the profitability of scale[Edit Line 216-217]. It has been modified by adding detailed explanations for SE, TE and PTE [Edit Line 154-158].

Reviewer 2 Report

One of the most important things in the empirical process of the DEA model is the selection of input and output variables. This is because, depending on which variable is selected, there is a big difference in the efficiency score value. In this paper, the source of the analysis data is based on survey and unstandardized interviews, and the subjective judgment of the researcher plays a lot in the selection and measurement of variables. Therefore, it is expected that supplementing the theoretical background and previous research that can support the variable selection process will help to strengthen the thesis of the study.

In addition, the number of references is small and there are many materials other than academic materials. If there are academic materials referenced in relation to the above theoretical background and prior research, reflecting them in the references will increase the reader's understanding of the paper.

Author Response

  • Additional explanations for variable selection have been added. [Edit Line 204-209]
  • References to previous studies have been added to improve the reader's understanding of the paper. [Edit Line 43-59, 163-183]
  • The explanation of the DEA analysis methodology and previous studies were revised to improve the understanding of the readers.
  • The contents of the introduction (especially the need for research) were revised. [Edit Line 43-59, 154-158, 163-183, 204-209]

Reviewer 3 Report

The manuscript makes a valiant attempt at assessing public R&D functions in South Korea, based on dated methods developed in 1957 and 1978. The manuscript is virtually impossible to read and understand by a non-expert audience, and has so many editorial inconsistencies and errors to make the task of the reader even more difficult. The authors should review the entire article for clarity and consistency, then consider submission again.

Author Response

  • The explanation of the DEA analysis methodology and previous studies were revised to improve the understanding of the readers.
  • The contents of the introduction (especially the need for research) were revised. [Edit Line 43-59, 154-158, 163-183, 204-209]
  • Additional explanations for variable selection have been added. [Edit Line 204-209]
  • References to previous studies have been added to improve the reader's understanding of the paper. [Edit Line 43-59, 163-183]

Round 2

Reviewer 1 Report

My comments have been addressed at a satisfactory level.

Author Response

Dear reviewer, thanks for your comments.

Reviewer 3 Report

If the editor and other referees believe the manuscript is now of sufficient quality to merit publication, I will follow their lead.

Author Response

Dear reviewer, thanks for your comments, we have revised our paper according to Academic Editor's comments.